# Her2 Expression in Circulating Tumor Cells Is Associated with Poor Outcomes in Patients with Metastatic Castration-Resistant Prostate Cancer

**DOI:** 10.3390/cancers13236014

**Published:** 2021-11-29

**Authors:** Denis Maillet, Nathalie Allioli, Julien Péron, Adriana Plesa, Myriam Decaussin-Petrucci, Sophie Tartas, Christophe Sajous, Alain Ruffion, Sébastien Crouzet, Gilles Freyer, Virginie Vlaeminck-Guillem

**Affiliations:** 1Service d’Oncologie Médicale, Institut de Cancérologie des Hospices Civils de Lyon, 69495 Pierre Bénite, France; julien.peron@chu-lyon.fr (J.P.); adriana.plesa@chu-lyon.fr (A.P.); myriam.decaussin-petrucci@chu-lyon.fr (M.D.-P.); sophie.tartas@chu-lyon.fr (S.T.); christophe.sajous@chu-lyon.fr (C.S.); gilles.freyer@chu-lyon.fr (G.F.); 2Centre de Recherche en Cancérologie de Lyon, INSERM 1052 CNRS UMR5286, Centre Léon Bérard, Université Claude Bernard Lyon 1, 69373 Lyon, France; virginie.vlaeminck-guillem@univ-lyon1.fr; 3Centre d’Innovation en Cancérologie de Lyon (CICLY), UR3738, Université Claude Bernard Lyon-1, 69921 Oullins, France; nathalie.allioli@univ-lyon1.fr (N.A.); alain.ruffion@chu-lyon.fr (A.R.); 4Institut des Sciences Pharmaceutiques et Biologiques, Faculté de Pharmacie, Université Claude Bernard Lyon1, 69008 Lyon, France; 5Service de Biostatistique et Bioinformatique, Hospices Civils de Lyon, 69495 Pierre Bénite, France; 6Laboratoire de Biométrie et Biologie Evolutive, Equipe Biostatistique-Santé, CNRS UMR 5558, Université Claude Bernard Lyon 1, 69622 Villeurbanne, France; 7Service d’Hématologie Biologique, Centre Hospitalier Lyon Sud, 69495 Pierre Bénite, France; 8Faculté de Médecine et de Maïeutique Lyon-Sud-Charles Mérieux, Université Claude Bernard Lyon 1, 69921 Oullins, France; 9Service d’Anatomie et de Cytologie Pathologiques, Centre Hospitalier Lyon Sud, Hospices Civils de Lyon, 69495 Pierre-Bénite, France; 10Service d’Urologie, Centre Hospitalier Lyon Sud, Hospices Civils de Lyon, 69495 Pierre-Bénite, France; 11Service d’Urologie, Hôpital Edouard Herriot, Hospices Civils de Lyon, 69003 Lyon, France; sebastien.crouzet@chu-lyon.fr; 12Faculté de Médecine Lyon Est, Université Claude Bernard Lyon 1, 69003 Lyon, France; 13Service de Biochimie Biologie Moléculaire Sud, Centre Hospitalier Lyon Sud, Hospices Civils de Lyon, 69495 Pierre-Bénite, France

**Keywords:** HER2 (ERBB2), androgen receptor splice variant 7 (AR-V7), highly sensitive assay, circulating tumors cells (CTCs), liquid profiling, metastatic castration resistant prostate cancer (mCRPC), androgen receptor signaling inhibitors (ARSI)

## Abstract

**Simple Summary:**

Although the role of HER2 in prostate cancer remains controversial, HER2 can be overexpressed during prostate cancer progression. In this study, we prospectively isolated circulating tumor cells (CTCs) from 41 men with metastatic castration-resistant prostate cancer (mCRPC). Then, we analyzed HER2 expression in CTCs with a highly sensitive assay. HER2 was frequently detected in CTCs of patients (*n* = 26, 63%) and we showed, for the first time, to our knowledge, that HER2 expression in CTCs was associated with poor long-term clinical outcomes in mCRPC. Moreover, the impact of HER2 expression in CTCs was an independent prognostic factor of progression-free survival.

**Abstract:**

HER2-dependent signaling may support the development of metastatic castration-resistant prostate cancer (mCRPC) by activating androgen receptor signaling through ligand-independent mechanisms. From 41 mCRPC patients (including 31 treated with Androgen Receptor Signaling Inhibitors [ARSI]), Circulating Tumor Cells (CTCs) were prospectively enriched with AdnaTest platform and analyzed with a multiplexed assay for *HER2* and *AR-V7* mRNA expression. Then, we evaluated the impact of HER2 expression on PSA-response, Progression Free Survival (PFS) and Overall Survival (OS). HER2 expression was detected in CTCs of 26 patients (63%). Although PSA response was similar regardless of HER2 status, patients with HER2 positive CTCs had shorter PSA-PFS (median: 6.2 months versus 13.0 months, *p* = 0.034) and radiological-PFS (6.8 months versus 25.6 months, *p* = 0.022) than patients without HER2 expression. HER2 expression was also associated with a shorter OS (22.7 months versus not reached, *p* = 0.05). In patients treated with ARSI, multivariate analyses revealed that the prognostic impact of HER2 status on PSA-PFS was independent of AR-V7 expression and of the detection of CTCs by an AdnaTest. We showed for the first time the poor prognostic value of HER2 expression in CTCs from patients with mCRPC. The therapeutic interest of targeting this actionable pathway remains to be explored.

## 1. Introduction

Because androgen receptor (AR) plays a critical role in the proliferation and migration of prostate cancer (PCa) cells, the vast majority of patients with hormone sensitive PCa (HSPC) respond to androgen-deprivation therapy (ADT). However, patients inevitably relapse despite castrate androgen levels, defined as castration resistance. Androgen receptor activity in castration-resistant PCa (CRPC) is partially restored through several mechanisms. Several androgen receptor signaling inhibitors (ARSI) have been recently developed to either suppress intratumoral androgen synthesis (CYP17A1 inhibitors such abiraterone) or block androgen activity (AR potent antagonists: enzalutamide, apalutamide or darolutamide). ARSI have dramatically improved patients′ outcomes, including overall survival (OS) in CRPC and metastatic HSPC patients [1]. Unfortunately, most of these patients develop mechanisms of resistance to ARSI, including AR reactivation, or activation of alternative cancer signaling pathways. ARSI resistance can be due to mutations of the ligand binding domain of AR, *AR* gene amplification, and the emergence of AR splice variants such as AR-V7 [2].

Pathways induced by tyrosine kinase receptors (TKRs) in the epidermal growth factor receptor (EGFR) family are potentially involved in CRPC progression. Indeed, upregulation of these TKRs is a major cause of PCa progression and relapse [3]. The molecular mechanism by which EGFR activity supports PCa progression remains unclear. The EGF TKR family consists of four members: EGFR (ERBB1, HER1), HER2 (ERBB2), HER3 (ERBB3) and HER4 (ERBB4). HER2 activation has been described to indirectly restore the androgen signaling pathway [4,5]. Several studies showed that HER2 expression is increased in a small subset of PCa tissue samples and that higher HER2 expression is associated with worse prognosis [6,7,8,9]. A recent study also reported that HER2 is overexpressed in CRPC bone metastases independently of gene amplification [10].

In the present study, we first evaluated *HER2* mRNA expression in circulating tumor cells (CTCs) from a cohort of metastatic CRPC (mCRPC) patients. Then, we prospectively assessed the impact of *HER2* mRNA expression in CTCs on PSA response and clinical outcomes (progression-free survival [PFS] and OS). Finally, we investigated the correlation between *HER2* and *AR-V7* mRNA expression.

## 2. Materials and Methods

### 2.1. Patients

We prospectively enrolled patients with progressive mCRPC as defined in the Prostate Cancer Clinical Trials Working Group 3 (PCWG3) guidelines [11]. Patients were required to have histologically confirmed prostate adenocarcinoma and progressive disease despite “castration level” of serum testosterone (<50 ng/DL) with continued ADT. Patients must have had documented metastases confirmed with a computed tomography (CT) and/or a technetium-99 bone scan. Patients with known neuroendocrine features or small cell prostate cancer were excluded. Prior treatments with ARSI, chemotherapy, or Radium-223 were allowed. Patients had to start chemotherapy or ARSI within 4 weeks after inclusion. All patients provided written informed consent. The ethical review board of Hospices Civils de Lyon approved the study (Board number: L16-160, 2016).

### 2.2. Study Design

In this study, we performed a multiplexed mRNA-based assay for the detection of HER2 and AR-V7 on enriched CTCs. Then, we prospectively assessed the impact of *HER2* mRNA expression on patient clinical outcomes in all patients and in the subgroup of patients treated with ARSI. A schematic representation of the study design is summarized in Appendix A. PSA was measured one month after treatment initiation and every 3 months thereafter. Chest, abdomen and pelvis CT scans were performed every 3–4 months and technetium-99 bone scans every 4–6 months. ARSI or chemotherapy were continued until progression or limiting toxicity. Results regarding the impact of AR-V7 status on PSA-PFS and radiological-PFS have been previously reported elsewhere [12].

### 2.3. CTC Isolation and Detection

The peripheral venous blood of patients was collected in collection tubes (BD vacutainer^®^ glass ACD solution B), which were immediately stored at 4 °C. The CTC enrichment procedure was started within 1–24 h after blood draw and performed using the AdnaTest ProstateCancerSelect kit (Qiagen, Hilden, Germany) as previously described. Briefly, CTCs were immunocaptured and magnetically separated from 5 mL whole blood using magnetic beads coated with a combination of antibodies against epithelial (EpCAM) and tumor-associated (HER2) antigens, followed by cell lysis. Then, messenger RNA (mRNA) was magnetically extracted from the cell lysate with oligo(dT)_25_-coated magnetic beads, using the AdnaTest ProstateCancerDetect kit (Qiagen, Hilden, Germany), in accordance with the manufacturer’s instructions. Next, CTC-specific mRNA was fully retrotranscribed using the SensiScript^®^ RT kit (Qiagen, Hilden, Germany). A 20-µL reaction volume was incubated for 60 min at 37 °C, followed by an inactivation step at 93 °C for 5 min. An expression analysis of three prostate cancer-associated transcripts (*PSA*, *PSMA* and *EGFR* genes) was performed by a multiplexed PCR reaction with 4 µL of template CTC-specific cDNA, using the AdnaTest ProstateCancerDetect kit (Qiagen, Hilden, Germany). The PCR reaction generated fragments with lengths of 449, 357, 163, and 120 bp, respectively, for PSMA, PSA, EGFR, and ACTIN (internal control). The PCR signal was detected by microcapillary electrophoresis with the Bioanalyzer system (Agilent Technologies, Santa Clara, CA, USA). A sample was considered positive—suggesting the presence of CTCs—if at least one peak related to these amplicons indicated a concentration of ≥0.10 ng/µL [12,13].

In parallel, the peripheral venous blood of patients was collected in CellSave^®^ tubes (Veridex^®^) to perform a CTC count, using the validated CellSearch^®^ system, within 24–96 h after blood draw. Immunomagnetic capture of CTCs was performed with antibody-coated beads directed against epithelial cell adhesion molecule (EpCAM) and detection of CTCs with the CellSearch^®^ system was performed with antibodies against epithelial cells (anti-cytokeratins 8, 18 and 19).

### 2.4. CTC-Based HER2 and AR-V7 Detection

To enhance the detection sensitivity of targeted biomarkers (*HER2* and *AR-V7*), CTC-specific cDNA was first pre-amplified by multiplex polymerase chain reaction (PCR).

This pre-amplification step is carried out in a 50 µL reaction comprised of 25 µL MultiPlex PCR Plus MasterMix 2X (Qiagen, Hilden, Germany), 6.3 µL undiluted RT product and 5 µL pre-amplification primer pool (a gift from Qiagen, Hiden, Germany), allowing simultaneously PCR amplification of HER2 and AR-V7. Pre-amplification PCR settings were 95 °C for 5 min, then 16 cycles with each cycle at 95 °C for 20 s, 60 °C for 20 s, 72 °C for 3 min.

Pre-amplified PCR samples were then used to perform two specific real time quantitative PCR (QPCR) reactions, following the QuantiNova™ SYBR^®^ Green RT-PCR kit instructions (Qiagen, Hiden, Germany). Each specific amplification is carried out in a 10 µL reaction comprised of 5 µL 2× mix QuantiNova™ SYBR^®^ Green RT-PCR Master Mix (Qiagen, Hiden, Germany), 2 µL pre-amplified 1:10 diluted PCR samples and 2 µL AdnaPanel^®^ primer (Qiagen, Hiden, Germany), allowing specific amplification of *HER2* or *AR-V7*. Runs were executed using AriaMx thermocycler (Agilent, Santa Clara, CA, USA) and settings were 95 °C for 2 min, then 35 cycles with each cycle at 95 °C for 5 s, 60 °C for 10 s, 77 °C for 10 s, followed by a melting curve analysis (60 °C–95 °C). A detailed analysis of the melting curves was first performed to exclude signals resulting from primer-dimers or non-specific amplifications, by comparing melting points with the corresponding values in the positive control sample included in the kit and checking that melting temperatures obtained for each amplicon was similar (±2 °C) to the manufacturer’s reference temperatures (Qiagen, Hiden, Germany). Threshold settings were achieved on positive control that amplified all targets and data were analyzed through the thermocycler software. A test result is regarded as positive if DeltaCt, using “Cutoff[gene]-SampleCt[gene]” mathematical formula, was greater than 0. The cut-off values, obtained from the manufacturer, were calculated for each gene separately as the mean Ct value of 20 different healthy donor blood samples minus a correction factor (Qiagen, Hiden, Germany).

We performed HER2 detection in all patients with Adnatest-mediated CTC immunocapture, independently from CTC detection with the CellSearch^®^ system because the lack of detection does not mean the lack of CTC, as previously suggested [12].

### 2.5. Statistical Analyses

We analyzed the impact of HER2 status on clinical outcomes including PSA-response, PSA-PFS, radiological PFS (rPFS), and OS. PSA response was defined as a decline of >50% from baseline. Categorical data were compared using Fisher’s test. PSA progression was defined as a post-treatment PSA level increase of at least 50% and a PSA ≥ 2 ng/mL. Radiological progression was defined using the RECIST1.1 criteria for soft tissue metastases and the PCWG3 criteria for bone metastases. PSA-PFS, rPFS, and OS were assessed using the Kaplan-Meier method. Differences in survival time were analyzed using a standard log-rank test. We built multivariate Cox models including HER2 status, the presence of CTCs (detected with AdnaTest^®^ (Qiagen, Hiden, Germany), as described in Section 2.3 and Section 2.4), and AR-V7 status. All tests were two-sided and *p*-values below 0.05 were considered statistically significant. Statistical analyses were performed using R software version 3.2.2 (R Core Team, Vienna, Austria).

## 3. Results

### 3.1. Patients Characteristics According to HER2 Status

Between October 2016 and March 2018, the AdnaPanel^®^ assay was prospectively performed on 41 patients with mCRPC treated at IC-HCL (Lyon, France). Of the enrolled men, 31 (75%) were treated with ARSI including 14 patients treated with abiraterone and 17 patients treated with enzalutamide, 9 (22%) patients were treated with chemotherapy and one patient progressed rapidly before treatment initiation and was treated with best supportive care alone. A CTC detection assay was performed using the AdnaTest^®^ platform in all patients (*n* = 41) and a CTC counting was performed using the CellSearch^®^ system in 30 patients. CTC detection rates were 78% (*n* = 32/41) with the AdnaTest^®^ assay and 83% (*n* = 25/30) with the CellSearch^®^ system. Because three patients presented CTC detected by CellSearch^®^ but not by AdnaTest^®^, we performed *HER2* and *AR-V7* mRNA expression analysis in all patients.

Among the patients included (*n* = 41), 23 (56%) were AR-V7+ and 26 (63%) were HER2+. The median baseline CTCs number on CellSearch^®^ was similar for HER2+ patients (10 per 7.5 mL of blood) compared to HER2− patients (8 per 7.5 mL of blood). The rate of patients with CTCs detected on AdnaTest^®^ was of 88% (*n* = 23/26) for HER2+ patients compared to 60% (*n* = 9/15) for HER2− patients but this difference was not statistically significant (Wilcoxon test, *p* = 0.052). All three HER2+ patients without CTC on AdnaTest^®^ had an AR expression and two of them also had an AR-V7 expression detectable with our highly sensitive assay. Of note, only one HER2+ patient had no CTCs on either AdnaTest^®^ platform and CellSearch^®^ system but had an AR-V7 expression that probably confirmed the presence of CTCs. There was no significant association between the HER2 expression status and the AR-V7 expression status (*p* = 0.51). There was no evidence of difference in Gleason score (*p* = 0.75), previous local treatment (*p* = 0.73), presence of pain (*p* = 1.0), prior treatment with ARSI (*p* = 0.45), presence of visceral metastases (*p* = 0.70), bone metastases (*p* = 0.34), lymph node metastases (*p* = 1.0) or median baseline PSA (*p* = 0.27) regarding HER2 status. However, 54% of HER2+ patients (*n* = 14/26) had a time to castration resistance lower than one year compared to 7% of HER2− patients (*n* = 1/15; *p* = 0.003) (Table 1).

### 3.2. Clinical Outcomes According to HER2 Status

Clinical outcomes according to HER2 status are summarized in Table 2 and Table 3. There was no statistically significant impact of the HER2 status on the PSA response rate. In the subgroup of patients treated with ARSI, the PSA response rate was 74% (*n* = 14/19) in HER2+ patients, and 92% (*n* = 11/12) in the other patients, but this difference was not significant (Figure 1).

The median PSA-PFS was 6.2 months for HER2+ patients and 13 months for HER2− patients (*p* = 0.034) (Figure 2). In the subgroup of patients treated with ARSI, HER2+, patients also had shorter PSA-PFS compared to HER2− patients (median of 6.2 months versus 18.5 months; *p* = 0.009) (Figure 3). The median rPFS was also significantly shorter for HER2+ patients than HER2− patients in the entire cohort (median of 6.8 months versus 25.6 months; *p* = 0.022) (Figure 2). In the subgroup of patient treated with ARSI, median rPFS was not reached for HER2− patients, and 6.3 months for HER2+ patients (*p* = 0.006) (Figure 3).

HER2 expression was also associated with shorter OS in the entire cohort (median of 22.7 months versus not reached; *p* = 0.05) (Figure 2). In the subgroup of patients treated with ARSI, HER2 detection in CTCs was also associated with a non-significant trend for shorter OS (Figure 3).

### 3.3. Survival Outcomes According to the Detection of CTCs

CTCs count (<5 or ≥5 CTCs per 7.5 mL of blood) was not associated with PSA-PFS, rPFS, or OS. However, the median PSA-PFS was 6.2 months among patients with detectable CTCs on AdnaTest^®^ platform compared to 23 months in patients with no detectable CTCs (HR = 3.0; 95% CI, 1.2–7.4; log-rank *p* = 0.016). The median rPFS was not reached in patients with no CTCs detectable and was of 6.77 months in patients with CTCs (HR = 3.4; 95% CI, 1.2–9.7; log-rank *p* = 0.016). Finally, OS was also shorter in patients with CTCs (median of 21.9 months versus not reached, *p* = 0.016).

### 3.4. Adjusted Analyses

In patients treated with ARSI, analyses adjusted on AR-V7 status and on the detection of CTCs by AdnaTest^®^ approach revealed that HER2 status was an independent prognostic factor of PSA-PFS and rPFS (Appendix A).

## 4. Discussion

To our knowledge, this is the first study that showed association of HER2 expression in CTCs with poor long-term clinical outcomes in a cohort of patients with mCRPC. We performed mRNA-based HER2 detection on enriched CTCs using AdnaTest^®^ platform with a pre-amplification multiplex PCR step followed by a specific analysis by qPCR to increased detection rate, as we already described for the detection of AR-V7 [12].

Unlike many other cancers such as breast or gastric cancers, it was previously described that PCa cells have a low HER2 expression [6,7,8,9]. Thus, Minner et al. described that a detectable HER2 immuno-staining on primary tumor was observed in 17.5% and 22.5% (according to the antibodies used) of cases in a cohort of more than 2000 localized PCa, but the large majority of samples had only 1+ staining, indeed a 2+ or 3+ staining was found in only 1.6% of samples included [8]. By contrast, we found that 63% of PCa CTCs present detectable *HER2* mRNA. This high detection rate could be explained by the highly sensitive method we used for *HER2* mRNA detection [12]. Moreover, it mostly results from the advanced cancer stage of our patients. This is consistent with the higher HER2 expression detection in 53% of mCRPC bone metastases described by Day et al. [10]. Moreover, many of our patients previously received ARSI, which could be correlated with HER2 expression in CTCs in previous studies [14].

Because some patients had CTCs detected on the AdnaTest^®^ platform but not with the CellSearch^®^ system and inversely, we decided to perform HER2 detection in all patients, even on those without CTC detected. Only one HER2+ patient had no CTC detected by AdnaTest^®^ and CellSearch^®^ approaches although showing AR-V7 expression. Note that assay used for AR-V7 detection was a highly sensitive assay, (due to a PCR pre-amplification step before qPCR reaction), while a classical PCR was used for AdnaTest^®^ CTC detection through at least one prostate cancer-associated transcript (PSA, PSMA, or EGFR) expression. HER2 and/or AR-V7 expression in patients showing no CTC detectable with AdnaTest^®^ or CellSearch^®^ approaches might be explained by a low number of CTCs isolated (proven by AR and AR-V7 expression not detected by classical PCR [data not shown] but detected with highly sensitive assays) and the presence of undifferentiated CTCs lacking biomarker surface expression (PSA, PSMA and EGFR) needed for CTC positivity on AdnaTest^®^ assay.

Few studies assessed *HER2* mRNA expression in CTCs from patients with PCa. One of them is the O’Hara study, where *HER2* mRNA analysis was not possible because of the background expression of *HER2* mRNA in leucocytes [15]; three other studies found HER2 positivity rates of 23% to 54% using real-time qPCR in CTCs of patients with metastatic PCa [15,16,17,18]. One of them showed, in a cohort of 42 patients, that *HER2* mRNA expression in CTCs was more frequently detectable in a metastatic setting (6 out of 11 patients) than in patients with localized PCa (3 out of 31) [16]. More recently, Josefsson et al. also reported a high 50% HER2 detection rate with a similar method to ours (PCR pre-amplification followed by qPCR) in CTCs of a cohort of 22 patients with PCa bone metastases (7 mHSPC and 15 mCRPC) [17]. Interestingly, the *HER2* mRNA expression profile of CTC samples was correlated with the expression profile of the corresponding metastatic tissues in most cases [17]. Finally, only one study published by Fantinato et al. evaluated the prognostic impact of HER2 detection in the CTCs of 43 patients with metastatic PCa and showed a non-significant trend of shorter PSA-PFS in patients with HER2 positive CTCs [18]. Interestingly, 13% to 35% of patients with metastatic PCa have an elevated serum level of HER2 protein, which was also correlated with an increased risk of death in patients with mCRPC [19,20,21].

The role of HER2 in PCa progression remains unclear. Recent studies have suggested that CTCs may contain a large amount of cancer stem cells or exhibit similar characteristics. It has been shown that EGFR (HER1) and HER2 are required to maintain the integrity of PCa cells in circulation and to favor their growth at distant sites [22]. It was also observed that EGFR is required for PCa cells sphere formation in in vitro analysis [10].

HER2 overexpression in mCRPC cells might also restore AR activity in mCRPC, through the activation of PI3K-AKT or MAPK pathways. HER2 activation mediates a down regulation of PI3K signaling followed by an AKT activation [23,24,25]. AKT and AR share the same complex signaling networks in which each protein is able to regulate the others with crosstalk between the two pathways. AKT is able to induce a phosphorylation of AR at S210 that decreases AR-mediated apoptosis and contributes to PCa progression [25,26]. Moreover, AKT can bind and phosphorylate AR at S213, which increases AR ligand-binding and promotes AR activation and translocation to the nucleus [27]. HER2 can also activate the MAPK pathway and especially ERK1/2 [28,29]. Similarly, AR and ERK1/2 signaling pathways share a same cross-talk complex [30]. ERK1/2 can regulate AR transcription but can also phosphorylate AR and its coregulators, resulting in AR activation and translocation into the nucleus. Consequently, HER2 activation might be responsible for an activation of AR signaling through ligand-dependent but also independent mechanisms.

Unlike breast cancer, HER2 targeted therapies have not yielded clinical benefit in mCRPC. A potential reason for this might be that the majority of patients included in these cohorts were not selected according to HER2 expression status [31]. HER2 inhibition alone might also not be sufficient in PCa because other EGF RTK family members could be involved in PCa progression. Indeed, a pan-EGFR inhibitor such as Dacomitinib (a HER2, HER3 and EGFR inhibitor) had better results in in-vitro studies and could consequently be used in future clinical trials [32]. Moreover, HER2 targeted therapies in metastatic PCa were mainly used alone without chemotherapy or ARSI, while the combination with chemotherapy is key in the treatment of HER2+ breast cancers. In PCa, preclinical studies showed that the HER2 inhibitor lapatinib enhanced enzalutamide or abiraterone activity in ARSI-resistant cell lines [25,33]. Future studies investigating HER2−targeted therapies, alone or in combination in metastatic PCa and the selection of patients with HER2 expression might be crucial.

The present results must be interpreted with consideration of the limited number of enrolled patients. Despite this, our results clearly show that HER2 expression was associated with poor survival outcomes in a cohort of mCRPC patients. We could also discuss a technical pitfall since the use of anti-EPCAM and anti-HER2 antibodies for CTCs isolation may have resulted in an enriched HER2 positive CTCs subpopulation and could partially explain our high HER2 detection rate. However, the AdnaTest^®^ platform is a validated approach for CTC enrichment, previously used by Antonarakis et al. [13]. Despite this, our results may indicate that HER2− patients had better outcomes than HER2+ patients. Additional studies are needed to confirm the impact of HER2 detection in CTCs on clinical outcomes of patients with mCRPC.5. 

## 5. Conclusions

In our study, we showed that *HER2* mRNA expression in CTCs is associated with worse clinical outcomes in patients with mCRPC. Prognostic value of HER2 expression in CTCs of patients with mCRPC must be confirmed in other prospective studies. In PCa, HER2 might be a novel targetable marker and its detection in CTC might be used in future therapeutic investigations of HER2 or pan-EGFR inhibitors.

## Figures and Tables

**Figure 1 cancers-13-06014-f001:**
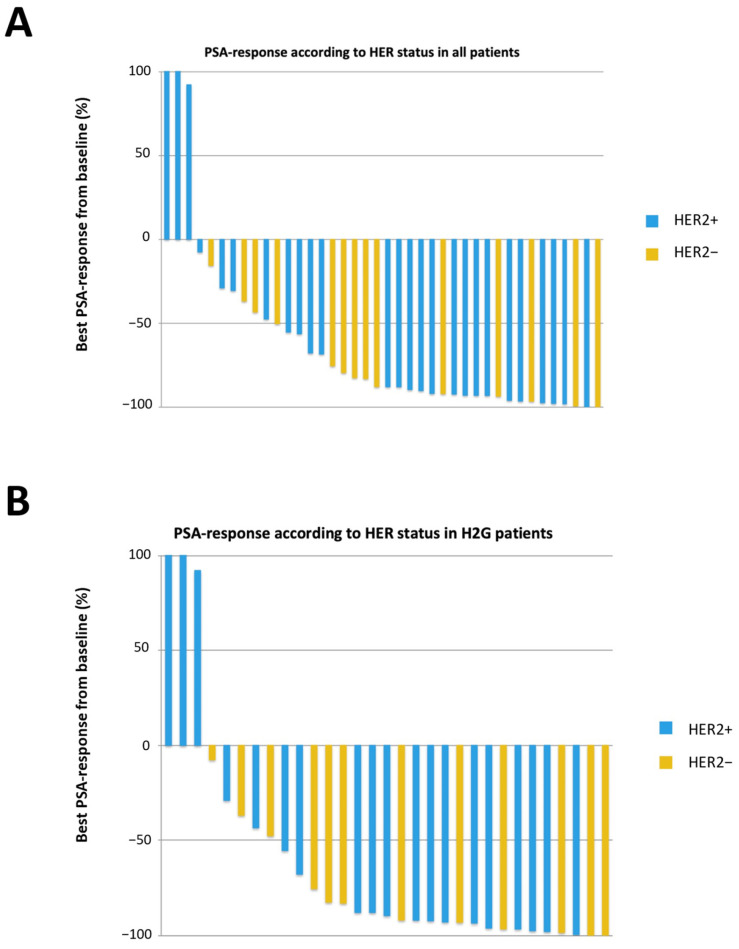
Waterfall plot depicting best prostate-specific antigen (PSA) response according to HER2 status (**A**) in all patients and (**B**) in patients treated with ARSI.

**Figure 2 cancers-13-06014-f002:**
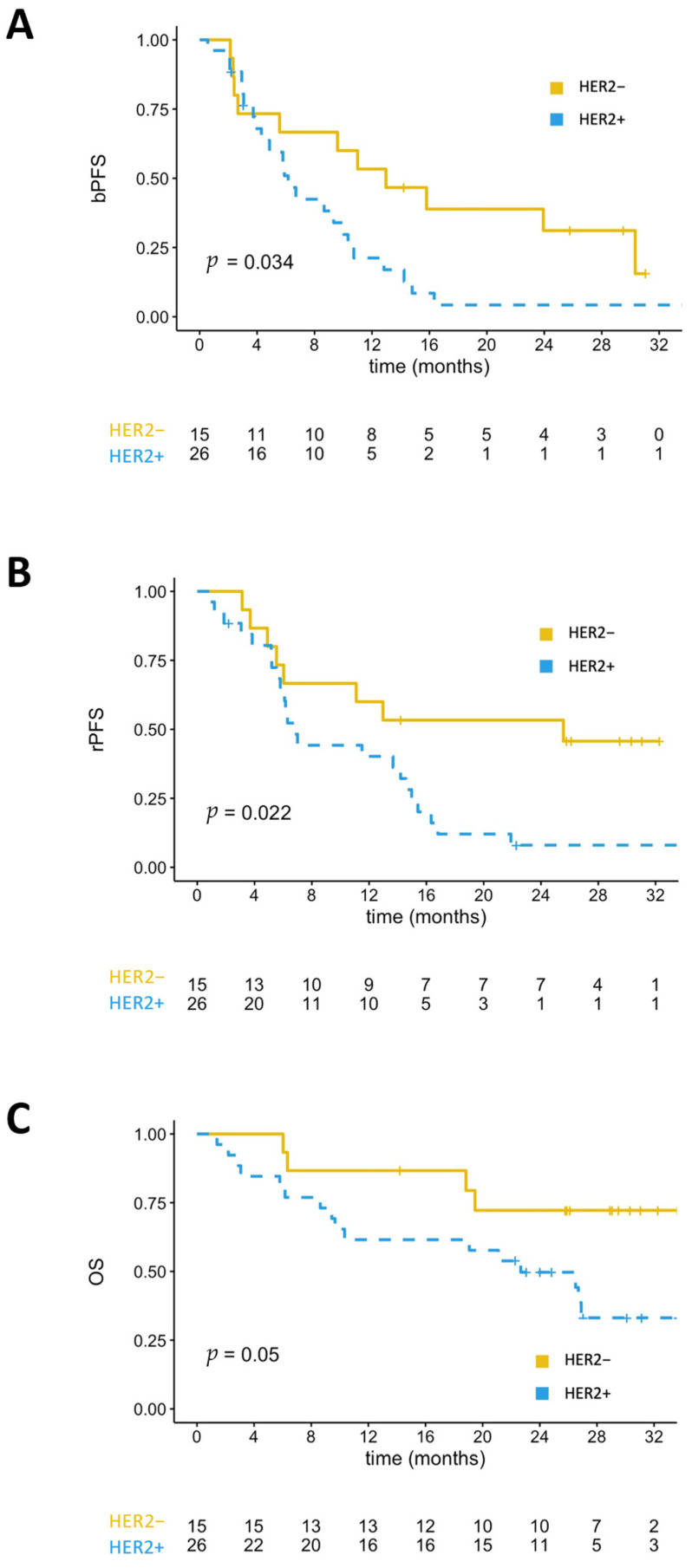
Kaplan-Meyer analyses in the whole cohort. (**A**) PSA-PFS of HER2+ versus HER2− patients. (**B**) Radiological-PFS (rPFS) of HER2+ versus HER2− patients. (**C**) Overall survival of HER2+ versus HER2− patients.

**Figure 3 cancers-13-06014-f003:**
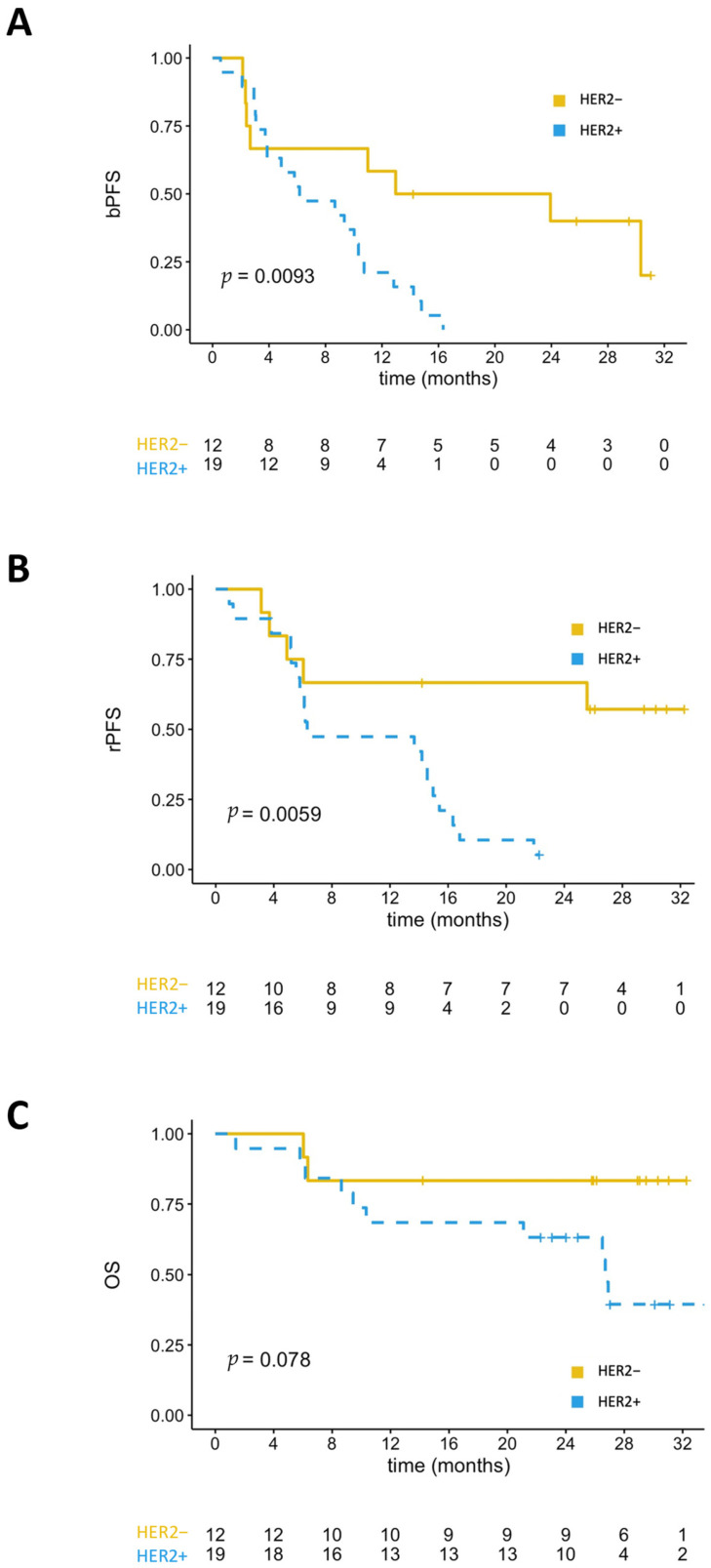
Kaplan-Meyer analyses in patients treated with ARSI. (**A**) PSA-PFS of HER2+ versus HER2− patients. (**B**) Radiological-PFS (rPFS) of HER2+ versus HER2− patients. (**C**) Overall survival of HER2+ versus HER2− patients.

**Table 1 cancers-13-06014-t001:** Patient characteristics according to CTC HER2 status.

Characteristic	CTC HER2+ (*n* = 26)	CTC HER2− (*n* = 15)	*p*-Value	Total (*n* = 41)
Median age (years)	76	74.7	0.27	73
Gleason sum ≥ 8 at diagnosis (%)	14 (54%)	9 (60%)	0.75	23 (56%)
M1 disease at PCa diagnosis (%)	9 (35%)	3 (20%)	0.45	12 (29%)
Previous local treatment (%) - Radical prostatectomy alone or with salvage or adjuvant radiotherapy - Radiotherapy alone - Other	17 (65%) 9 (35%) 7 (27%) 1 (4%)	11 (73%) 6 (40%) 5 (33%) 0 (0%)	0.73 - - -	28 (68%) 15 (37%) 12(29%) 1 (2%)
Time to castration resistance <1 year (%)	14 (54%)	1 (7%)	0.003	15 (37%)
Presence of pain (%)	11 (42%)	5 (33%)	1.0	16 (39%)
Prior treatment before subsequent treatment initiation (%) - 1 line - ≥2 lines	11 (42%) 4 (15%) 7 (27%)	5 (33%) 1 (7%) 4 (27%)	0.76	16 (39%) 5 (12%) 11 (27%)
Prior treatment with ARSI (%)	5 (19%)	5 (33%)	0.45	10 (24%)
Prior treatment with chemotherapy (%)	9 (35%)	4 (27%)	0.72	13 (32%)
Presence of visceral metastases (%)	5 (19%)	4 (27%)	0.70	9 (22%)
Presence of lymph node metastases (%)	11 (42%)	7 (47%)	1.0	18 (44%)
Presence of bone metastases (%)	24 (92%)	12 (80%)	0.34	36 (88%)
Median baseline PSA (ng/mL)	69.9	28.3	0.27	35
Median baseline CTC count on CellSearch (per 7.5 mL of blood)	10	8	0.82	9
Detection of CTCs (AdnaTest)	23 (88%)	9 (60%)	0.052	32 (78%)
Detection of AR-V7 (AdnaPanel)	16 (61%)	7 (47%)	0.51	23 (56%)
Type of subsequent therapy (%) - ARSI - Chemotherapy - None	19 (73%) 6 (23%) 1 (4%)	12 (80%) 3 (20%) 0 (0%)	-	31 (75%) 9 (22%) 1 (2%)

**Table 2 cancers-13-06014-t002:** Clinical outcomes according to HER2 status for all patients.

Clinical Outcome	CTC HER2+ (*n* = 26)	CTC HER2− (*n* = 15)	*p*-Value
PSA response	73%	73%	-
Median PSA-PFS (months)	6.2	13.0	0.034 * HR = 2.20 (95% CI, 1.0–4.6)
Median radiological-PFS (months)	6.8	25.6	0.022 * HR = 2.55 (95% CI, 1.1–5.9)
Median overall-survival (months)	22.7	NR	0.05 * HR = 2.85 (95% CI, 0.95–8.5)

* Log-rank test. NR: Not reached.

**Table 3 cancers-13-06014-t003:** Clinical outcomes according to HER2 status in the subgroup of patients treated with ARS.

Clinical Outcome	HER2+ (*n* = 19)	HER2− (*n* = 12)	*p*-Value
PSA response	74%	92%	-
Median PSA-PFS (months)	6.2	18.5	0.009 * HR = 3.35 (95% CI, 1.3–8.7)
Median radiological-PFS months)	6.3	NR	0.006 * HR = 4.26 (95% CI, 1.4–12.9)
Median overall-survival (months)	22.7	NR	0.078 * HR = 3.59 (95% CI, 0.8–16.4)

* Log-rank test. NR: Not reached.

## Data Availability

The data presented in this study are available on request from the corresponding author. The data are not publicly available due to ethical consideration related to the privacy of medical data of patients included in this study.

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
