# Peer review of "Her2 Expression in Circulating Tumor Cells Is Associated with Poor Outcomes in Patients with Metastatic Castration-Resistant Prostate Cancer"

_cancers, 2021, doi:10.3390/cancers13236014_

Round 1

Reviewer 1 Report

The study from Maillet and colleagues demonstrates the utility of Her2 as a prognostic biomarker in the CTCs of men with castration-resistant prostate cancer. It is my opinion that this study will be a valuable contribution to the field, however, there are several issues that would need to be addressed for the paper to be suitable for publication. Specifcally, the authors should provide more methodological detail and justification, so the readers can understand how the data was generated and the limitations of the study.

Specific Comments

  1. I find the following statement in the introduction confusing and it would benefit from further clarification/elaboration.
    1. “Therefore, unlike breast cancer, overexpression of HER2 protein may be sufficient to support PCa progression and metastasis development.”
  2. Although the methods state that the patients were selected according to PCWG3 guidelines, it would be helpful to include the key criteria for inclusion. Specifically, I would expect all these patients to have previously progressed on ADT to be considered CRPC?
  3. A follow on from comment 1, could the authors please clarify in their text what is considered prior treatment before subsequent treatment initiation in Table 1. I assume that this is for ARSI or Chemo, as all these patients should have had prior treatment with ADT?
  4. It would also be helpful to have a brief overview of how CTC enrichment is conducted with AdNa and CellSearch systems. Specifically, do they both use EpCam and Her2 for CTC enrichment? This is important due to the limitations of using this selection approach (as noted in the discussion).
  5. The authors should justify the use of HER2 in CTC enrichment. Is this a common approach for CTC selection, or was the biomarker added to increase detection of HER2+ cells?
  6. What is the rationale for including all 41 patients in the downstream analysis when only 35 had detectable CTCs? Wouldn’t the samples without CTC detection result in potential false negatives or false positives (if the enriched samples had residual leucocytes), for HER2? Is there a precedent from other studies to include samples even if CTCs are not detected, and if so, what is the explanation for this?
  7. A subset of men with CRPC can undergo neuroendocrine transdifferentiation as they develop resistance to therapy. Do the authors know if any patients have neuroendocrine disease, and if so, the HER2 status in this subset? If the neuroendocrine status is not known, this should be noted in the manuscript.

Author Response

We would like to thank the reviewer#1 for its thorough review and helpful comments. Concerns raised are addressed below in the attached document. 

Reviewer 2 Report

The paper is interesting and worth publishing. Several issues remain to be elucidated though.

  1. The prognosis of pts with and without HER2 expression differs substantially. At the same time no information on previous local management is provided. Were the pts subjected to prostate surgery or radiotherapy? Was the previous treatment in analyzed cohorts comparable?
  2. What was the pattern of prostate cancer dissemination? Is it possible to provide data on lymph nodes, bones as well as other organs  involvement?   

Author Response

We would like to thank the reviewer#2 for its thorough review and helpful comments. Concerns raised are addressed below in the attached document. 

Reviewer 3 Report

The manuscript by Maillet et al described the poor prognostic value of HER2 expression in CTCs from patients with mestastatic castration resistant prostate cancer. The manuscript is very interesting, well-written and clear. The study design is linear. To further improve the manuscript, I only suggest to add a schematic representation of the experimental approach.

Author Response

We would like to thank the reviewer#3 for its thorough review and helpful comments. Concerns raised are addressed below in the attached document. 
